# Epitranscriptomics in the Glioma Context: A Brief Overview

**DOI:** 10.3390/cancers17040578

**Published:** 2025-02-08

**Authors:** Pablo Santamarina-Ojeda, Agustín F. Fernández, Mario F. Fraga

**Affiliations:** 1Foundation for Biomedical Research and Innovation in Asturias (FINBA), 33011 Oviedo, Spain; pablosoov76@gmail.com (P.S.-O.); agustin.fernandez@cinn.es (A.F.F.); 2Health Research Institute of Asturias (ISPA), 33011 Oviedo, Spain; 3Nanomaterials and Nanotechnology Research Centre (CINN-CSIC), 33940 El Entrego, Spain; 4Institute of Oncology of Asturias (IUOPA), University of Oviedo, 33006 Oviedo, Spain; 5Centre for Biomedical Network Research on Rare Diseases (CIBERER), 28029 Madrid, Spain

**Keywords:** RNA, mRNA, epitranscriptomics, glioma, methylation

## Abstract

Epitranscriptomic modifications impact RNA stability and function, influencing key biological processes. While initially behind epigenetics, the field has advanced in the last decade, linking these modifications to development and diseases like cancer. High-grade gliomas, such as glioblastoma, remain challenging due to their aggressiveness and lack of effective treatments. Understanding the impact of epitranscriptomic alterations in gliomas may provide new insights into tumor progression and therapeutic strategies. This review provides an overview of RNA modifications in tRNAs, rRNAs, miRNAs, and mRNAs, exploring the role of the proteins involved in epitranscriptomics and their dysregulation in gliomas. We discuss how this imbalance affects pathways in glioma progression and highlight recent advancements in RNA modification-based therapeutics.

## 1. Introduction

Epigenetics is essential for cell regulation and influences processes ranging from development to aging and environmental adaptation [1]. In this regard, epitranscriptomics, the study of RNA modifications, has gained increasing significance alongside established DNA and histone mechanisms [2,3,4]. Post-transcriptional RNA modifications are found across various RNA types, including transfer RNA (tRNAs), ribosomal RNA (rRNA), messenger RNA (mRNA), and other non-coding RNAs (ncRNAs), forming a complex layer of regulatory control [5]. Although some modifications, like pseudouridine (Ψ), inosine (I), and dihydrouridine (D), were recognized decades ago in tRNAs [6], research has since identified more than 150 modifications across all RNA species [7,8]. RNA modifications play essential roles across processes such as transcription, splicing, stability, translation, and decay, which collectively are critical for proper cellular function [9,10,11]. Similar to DNA and histone modifications, RNA marks are sensitive to environmental cues, actively shaping the epitranscriptome of an organism [12].

Evolution has favored a wide range of RNA modifications that support their specific functions. A comprehensive review of the wide variety of RNA modifications was published by McCown and colleagues [8], and extensive information concerning RNA modification dynamics can be found in the MODOMICS database [7]. The dynamic behavior of epitranscriptomic marks was hypothesized prior to identifying the first demethylase acting on N^6^-methyladenosine (m^6^A) in mRNA [13,14,15], inspiring further research into the description, mapping, and roles of these RNA marks. The epitranscriptome is shaped by internal and external signals through the activity of mark-specific proteins. Analogous to DNA and histone chemical modifications, equivalent terms are used in epitranscriptomics to refer to the distinct players involved in its regulation. The term “writers” refers to proteins or complexes that catalyze the modification of RNA nucleotides, converting them into distinct chemical entities. The term “erasers” denotes proteins that remove these modifications from RNA. Finally, the “readers” are proteins that bind to RNA modifications and trigger downstream signaling pathways. The high specificity of proteins involved in these processes is particularly noteworthy. Considering that natural selection has maintained this entire regulatory network, its importance is unquestionable.

In this review, we discuss epitranscriptomic modifications across various RNA species, including tRNAs, rRNAs, mRNAs, and other ncRNAs (Figure 1). We provide an up-to-date overview of key regulatory factors for each chemical modification and explore the relationship between RNA modifications and glioma tumorigenesis (Figure 2, Table 1). Additionally, we highlight potential therapeutic strategies targeting RNA-modifying proteins, as well as the challenges that may arise. A key point of this review is the inclusion of the most recent studies in epitranscriptomics, a rapidly evolving field. This ensures that the latest advancements are reflected and an updated perspective on the current understanding of epitranscriptomics in gliomagenesis is provided. In addition, numerous studies over the past decade have highlighted the growing interest in understanding the role of RNA modifications in glioma biology.

## 2. Description of the Key RNA Modifications

### 2.1. Chemical Modifications in tRNAs

tRNAs are cloverleaf-shaped RNA molecules that translate mRNA information into amino acid chains for protein synthesis [16]. They undergo over 50 distinct post-transcriptional modifications, accounting for approximately 80% of all modifications reported in all RNA species [7]. tRNA modifications are critical for function and stability. Modifications in the anticodon region, especially at positions 34 and 37, are essential for accurate protein synthesis, with the wobble base at position 34 allowing for flexible base pairing to recognize additional codons [17,18]. Structural modifications in the main body of tRNAs support higher-order configurations through D- and T-loop interactions [19]. Modifications at position 37, like N^6^-threonylcarbamoyladenosine (t^6^A), stabilize tRNAs and prevent frameshift mutations during translation [20,21]. While anticodon modifications have been extensively studied, research on the R- and L-loops has shown their role in tRNA stability and ribosome translocation, impacting translation efficiency [22]. tRNA methylation, mediated by NSun2, NSun4, and Dnm2, regulates stability by controlling angiogenin-induced cleavage, and deletion of these enzymes leads to cleaved tRNA fragments linked to neurological defects in mice [23]. The evolutionary conservation of methyltransferases highlights the importance of tRNA methylation, since alterations in its patterns can have physiological consequences. For example, the TRM6/TRM61A complex catalyzes m^1^A methylation at position 58 of the T-loop in initiator methionine tRNAs, stabilizing its structure [24], and dysregulation of the TRM6/TRM61A complex has been implicated in various tumors, including gliomas, by altering translation patterns [25,26].

### 2.2. Chemical Modifications in rRNAs

rRNAs are essential non-coding RNAs that constitute the structural and functional core of ribosomes, representing the most abundant RNA class in eukaryotic cells [27]. In humans, ribosomes contain four rRNAs—5S, 18S, 5.8S, and 28S—that contribute to ribosome structure and catalyze peptidyl transferase reactions essential for protein synthesis [28]. Evolutionarily linked to ribosomes, rRNAs have specific nucleotide modifications that regulate their function. These modifications include 2′-O-methylation (Nm), m^6^A, and pseudouridylation, positioned at key sites for stability and function during translation [29,30]. Pseudouridine marks are deposited by pseudouridine synthases, which utilize H/ACA small nucleolar RNAs (snoRNAs) as guides to incorporate these modifications into RNAs [31]. The loss of pseudouridine can disrupt ribosome recruitment and reading frame accuracy, since pseudouridine is crucial for supporting rRNA 3D structure [32]. Pseudouridylation is also related to oncogenesis, since mutations in the *DKC1* gene, which encodes a protein involved in H/ACA snoRNA biosynthesis, suggest a tumor suppressor role, with alterations reported in pituitary adenoma and hepatocellular carcinoma [33,34,35]. snoRNAs also guide O-methylation, which occurs early in ribosome assembly within the nucleolus [36]. Mutations affecting rRNA methylation are linked to human diseases, with O-methylation being essential for protein translation and influencing physiological processes such as neurodevelopment [37,38,39].

### 2.3. Chemical Modifications in Other ncRNAs

Non-coding RNAs, which are not translated into proteins, play key roles in regulating processes from transcription to translation. microRNAs (miRNAs), 20–22 nucleotides long, are generated from precursors pri-miRNAs [40], with a critical role in mRNA dynamics. Processing and activity are modulated by chemical modifications. m^6^A regulates miRNA processing by aiding the recognition of the machinery for maturation [41]. Other modifications, like O-methylation, prevent miRNA degradation by PNPase 1 [42], while 7-methylguanosine (m^7^G) affects miRNA maturation and activity, particularly in lung cancer [43]. A-to-I editing, a modification converting adenosine to inosine, impacts miRNA processing by disrupting the Drosha/Dicer machinery and altering mRNA targeting [44]. This editing is crucial for processes like neurodevelopment [45] and is linked to diseases such as cancer, neurological disorders, and cardiopathies [46]. Long non-coding RNAs (lncRNAs), longer than 200 nucleotides, are also involved in genome dynamics, cell structure, and gene expression [3]. lncRNAs such as *XIST* mediate X chromosome repression through m^6^A modifications deposited in the *XIST* sequence, which are recognized by RNA-binding proteins [47,48]. Similarly, m^6^A modifications in lncRNAs like *NORAD* influence cellular senescence in intervertebral disc cells [49], while m^6^A patterns in lncRNA *Dubr* support neurodevelopment by stabilizing the YTHDF1/3 complex, thereby facilitating mRNA translation [50].

### 2.4. Chemical Modifications in mRNAs

Modifications in mRNAs have gained attention due to their direct impact on protein synthesis. Among them, one of the most studied is m^6^A, the first reversible mRNA modification to be described [51,52]. The roles of m^6^A in mRNA are well established, particularly in mRNA stabilization and the recruitment of RNA-binding proteins. It regulates key processes such as mRNA maturation, translation, and decay, which makes m^6^A essential for gene expression and protein synthesis [11]. Pseudouridine in mRNA is linked to read-through at stop codons and pre-mRNA splicing [53,54]. mRNA capping involves the methylation of guanosine at the 5′-end to form m^7^G, which protects mRNA from degradation and ensures proper splicing and translation [55]. In addition, A-to-I editing is a common modification in mRNAs that enhances their stability by modulating miRNA interactions and increases protein functional diversity through changes in protein sequence [56,57]. 5-methylcytosine (m^5^C) is also present in mRNA, and its functions have been associated with mRNA export and stabilization [58].

In summary, RNA modifications are crucial for regulating RNA stability, maturation, and translation. While some of them may appear redundant, each modification likely operates through distinct signaling pathways and mechanisms and contributes to different functions.

## 3. Role of RNA Modifications in Gliomagenesis

Epitranscriptomic patterns regulate neurodevelopment and CNS function, influencing cell homeostasis through a regulatory network including writer, eraser, and reader proteins [59]. Other reviews have addressed and discussed the huge influence of epitranscriptomic alterations in the onset of neurodevelopmental and neurological disorders, neurodegenerative diseases and in the pathogenesis of brain tumors [60,61,62]. Brain tumors arise from the central nervous system (CNS) and spinal cord, exhibiting a diverse range of origins, mutations, and developmental trajectories. Gliomas, tumors apparently originating from glial cells, account for 23% of all tumors, with glioblastoma (GBM) being the most common malignant tumor associated with a poor prognosis [63]. In adults, gliomas are classified based on molecular markers that influence patient prognosis: *IDH*-mutant/1p19q-codeleted oligodendrogliomas, with the best prognosis; *IDH*-mutant/1p/19q-non-codeleted astrocytomas, with an intermediate prognosis; and *IDH*-wildtype astrocytomas (glioblastoma), with the poorest prognosis [64]. The brain is characterized by cellular heterogeneity, high functional plasticity, and dependence on precise gene regulation. These features make it uniquely susceptible to epigenetic and epitranscriptomic alterations, driving interest in studying these mechanisms in it rather than in other organs. Despite epigenetic and epitranscriptomic mechanisms being studied in some types of tumors, most published reports regard brain and hematological malignancies, where mutations or alterations in epigenetic-related proteins such as IDH1, DNMT1, and TET2 are commonly observed [64,65]. In brain tumors, genetic and epigenetic imbalances critically disrupt cellular homeostasis. However, genetic alterations are not as predominant as in other tumors, making the study of epigenetic/epitranscriptomic regulation essential [66]. Efforts have been made to deconvolve genomic and epigenomic complexity, but the role of epitranscriptomics remains less explored. The following paragraphs will discuss the role of different RNA modifications in glioma biology (also see Table 1).

m^6^A is closely associated with brain neoplasms, primarily through reader proteins that bind to m^6^A-modified mRNA, such as YTHDF1-3 and YTHDC1-2. This association is further strengthened by the frequent dysregulation of m^6^A writer complexes—METTL3, METTL14, and the METTLE3 adapter WTAP—and erasers—FTO and ALKBH5—in this type of tumors [67,68,69,70]. Studies in glioma stem cells (GSCs) revealed an enrichment in oncogenic m^6^A-marked transcripts, primarily driven by *YTHDF2* upregulation [71,72,73]. Although YTHDF2 facilitates m^6^A-dependent mRNA decay, it selectively promotes the stability of specific oncogenic transcripts, such as *MYC* and *VEGFA*, in GSCs [72]. In contrast, it accelerates the degradation of *LXRA* and *HIVEP2* mRNAs, which are typically downregulated in GBM. Notably, YTHDF2 levels are further increased through phosphorylation, enhancing signaling pathways [74] (Figure 2A). m^6^A also contributes to brain tumorigenesis by regulating other RNA species and their associated regulators. For instance, *LINC00839*, a lncRNA related to stemness and radiotherapy resistance, is regulated by the m^6^A signaling pathway. In GSCs, m^6^A methyltransferase METTL3 is upregulated and directly targets *LINC00839* to methylate different nucleotides, which will be subsequently recognized by YTHDF2 to avoid *LINC00839* degradation. Thus, the stability and function of *LINC00839* are supported through the METTL3/YTHDF2 interaction, ensuring its activity within the cell [70,75]. METTL3 also deposits m^6^A on transcripts encoding splicing factors, influencing alternative splicing and driving the production of oncogenic mRNAs, such as *BCL-XS* and *NCOR2* [76]. *METTL3* expression is enhanced by pathways commonly dysregulated in this tumor, such as PDGF-EGR1. GSCs with an activated PDGF/EGR1 pathway exhibit elevated METTL3 levels, resulting in hypermethylation of *OPTN* and its subsequent degradation. As a negative regulator of GBM growth, OPTN suppresses tumor progression both in vitro and in vivo, with reduced OPTN levels being directly associated with increased GSC proliferation and self-renewal [77].

m^6^A erasers, ALKBH5 in particular, play a similarly crucial role in the regulatory network. ALKBH5 expression is elevated in GSCs and correlates with poor prognosis. Its activity is modulated by the lncRNA *LOC100507424*, which facilitates its interaction with *FOXM1* transcripts, enhancing *FOXM1* expression and promoting tumorigenesis in GSCs [78]. In *IDH*-mutant gliomas, slower tumor growth and better prognosis as compared to their *IDH*-wildtype counterparts are attributed to reduced FTO levels, another m^6^A demethylase. The oncometabolite 2-hydroxyglutarate (2-HG) produced by the *IDH* mutation disrupts the function of demethylases like FTO, altering the epigenetic landscape of these tumors [79]. A recent study demonstrated that pharmacological inhibition of FTO in *IDH*-wildtype gliomaspheres implanted into mouse brains mimicked the less aggressive phenotype of *IDH*-mutant tumors. Inhibiting FTO led to m^6^A hypermethylation and suppression of the *ATF5* transcript, a gene that supports cell survival [80,81]. GSCs modify their metabolism to create a tumor-permissive environment, which in turn affects epitranscriptomic modifications [82]. Genetic and pharmacological targeting of malate dehydrogenase (MDH2) increases m^6^A levels in transcripts like *PDGFR-β*, promoting GSC stemness and contributing to the tumoral phenotype. Reduced MDH2 expression increases alpha-ketoglutarate (αKG) levels, enhancing the demethylation of m^6^A in the *PDGFR-β* transcript, reverting the previous phenotype [82,83]. Gliomas are characterized by mutations in some specific genes, including *EGFR*, *TERT*, or *TP53* among others, resulting in altered pathways that contribute to tumorigenesis through uncontrolled cell proliferation and survival [63]. Alterations in some of these commonly dysregulated genes have been associated with disruptions in m^6^A methylation. Specifically, EGFR activation has been shown to directly reduce m^6^A levels by inhibiting the nuclear export of ALKBH5 [84]. Moreover, *YTHDF2* overexpression is linked to mutated *TP53*, hampering the expression of some tumor suppressor genes, including *CDKN2B*, commonly lost in GBM patients [85]. *MGMT* activity in repairing DNA damage caused by temozolomide treatment has long been recognized as a biomarker for patient response to this therapy. While *MGMT* promoter DNA methylation is a key regulator, m^6^A modifications also play a crucial role in stabilizing *MGMT* mRNA. Experiments conducted on differentiated GSCs revealed that *METTL3* downregulation leads to reduced m^6^A levels. This reduction compromises *MGMT* mRNA stability, ultimately enhancing the sensitivity of GSCs to temozolomide [86].

Pseudouridine distribution has been investigated for its potential links to neurological diseases. As previously mentioned, this modification is introduced through two mechanisms: the H/ACA box small nucleolar RNA-dependent pathway and the pathway involving pseudouridine synthase (PUS), which catalyzes the isomerization of uridine [87]. PUS7, a pseudouridine writer in tRNAs, is often upregulated in GSC, where it enhances cell growth and tumorigenesis by modulating the TYK2-IFN pathway through tRNA pseudouridylation [88] (Figure 2B). Although research on this topic is still limited, some studies suggest that other pseudouridine synthase genes may be related to prognosis in low-grade gliomas [89], with evidence connecting their expression to tumor growth [90]. DKC1 upregulation, another pseudouridine writer, enhances small RNA stability and telomerase activity in hematopoietic stem cells and lung cancer, and its knockdown reduces N-cadherin, HIF-α, and MMP2 levels in human glioma cell lines [90].

Regulatory factors involved in m^5^C deposition on RNA species have been linked to brain tumorigenesis. Elevated m^5^C levels have been observed in glioma tissue compared to non-tumor brain tissue, with these levels inversely correlated with patient prognosis [91]. A recent example of this factors is NSUN4, an m^5^C writer that catalyzes methylation in various RNA molecules, including *CDC42* mRNA. m^5^C at *CDC42* promotes its interaction with the m^5^C reader ALYREF, stabilizing the mRNA and enhancing translation. This leads to increased CDC42 protein levels, which activate the PI3K-AKT signaling pathway and drive glioma progression in vitro and in vivo [92]. Another m^5^C writer, NSUN2, is involved in the dysregulation of *VEGFR2* in GBM endothelial cells by adding m^5^C to the lncRNA *LINC00324*, thereby stabilizing it. *LINC00324* is a target of AUH, a protein that inhibits CBX3 translation. The loss of this inhibition stabilizes *CBX3* mRNA, ultimately enhancing *VEGFR2* expression and promoting angiogenesis [93] (Figure 2C). RNA dynamics involving both mRNA and lncRNA are frequently observed in gliomas. For instance, a study by Wu and colleagues describes how the m^5^C and 5-hydroxymethylcytosine (hm^5^C) writers, NSUN5 and TET2, respectively, mediate the methylation of the chromatin-associated RNA *CTNNB1* at the fifth cytosine position, followed by its oxidation to hm^5^C, promoting *CTNNB1* degradation through RBFOX2 and subsequent phagocytosis by tumor-associated macrophages (TAMs) [94]. Furthermore, *NSUN5* expression is modulated by DNA methylation, which introduces additional complexity to this pathway, and is negatively correlated with *IDH* mutations. This example highlights the complexity of interpreting epitranscriptomic data in tumors, where multiple regulatory mechanisms interact, and epigenetic and epitranscriptomic alterations can synergistically drive tumorigenic phenotypes. Notably, the downregulation of *NSUN5* in gliomas, resulting from CpG island hypermethylation at its promoter region, affects another key epitranscriptomic mechanism: the absence of NSUN5 leads to hypomethylation (m^5^C) at position C3782 of human 28S rRNA. This hypomethylation disrupts the three-dimensional stability of the translational complex—comprising rRNA, tRNA, and mRNA—thereby contributing to tumorigenesis, as demonstrated in glioma models in vitro and in vivo [95]. Other RNA methylation mechanisms impacting glioma are mediated through tRNAs at m^1^A positions. Disruptions in the TRM6/61 complex, responsible for adding m^1^A at the 58th position of initiator methionine tRNA, can disrupt the balance of this modification and potentially promote glioma cell proliferation if not tightly regulated [25,26].

A-to-I editing can impact various RNA species through specific editors, such as ADAR1/2/3. High expression of *ADAR1* has been observed in GSCs, where it increases the inosine content in GM2A mRNA, a key regulator of ganglioside catabolism, promoting stemness [96]. Notably, *ADAR1* expression may be upregulated in GBM by METTL3-mediated mRNA methylation, followed by stabilization via YTHDF1 [97]. Distinct A-to-I editing patterns in glioma compared to normal tissues offer potential for patient classification, as well as for improving diagnosis and prognosis and identifying tumor vulnerabilities [98]. This editing mechanism affects not only mRNAs but also miRNAs in gliomas. Altered A-to-I editing events in various miRNAs under non-physiological conditions can have deleterious effects, as seen with miR-589-3P. This molecule displays a different A-to-I editing profile in GBM compared to non-tumor brain tissues, a reaction mediated by the adenosine deaminase *ADAR2* [99]. Unedited miR-589-3P acts as an oncogenic-miRNA, enhancing cell proliferation and invasion. The shift in miRNA targeting driven by changes in the epitranscriptomic landscape is of particular interest. In non-tumor tissues, edited miR-589-3p targets the *PCDH9* transcript, promoting its protein synthesis and correlating with higher *ADAR2* levels. In contrast, GBM exhibits low *ADAR2* expression, leading to reduced editing of miR-589-3P, which then shifts its target to *ADAM12*, which encodes a metalloproteinase that drives GBM proliferation [99] (Figure 2D). A similar effect was reported for miR-376, as reduced A-to-I editing in GBM patients leads to decreased *RAP2A* and increased *AMFR* expression, ultimately promoting cell migration and invasion [100].

**Table 1 cancers-17-00578-t001:** List of some RNA modifications and their effect on gliomagenesis.

RNA Modification	Behavior in Disease	Effector Involved	Genetic Target	Model	Reference
m^1^A	m^1^A increase	TRM6/61	Up- Methionine tRNA	Rat glioma cells	[25]
m^6^A	m^6^A increase	YTHDF2	Upregulation of (Up-) *MYC*, *VEGF*	GSCs	[72]
	m^6^A increase	YTHDF2	Downregulation of (Down-) *LXRA*, *HIVEP2*	GSCs	[74]
	m^6^A increase	METTL3, YTHDF2	Up- *LINC00839*	GSCs	[75]
	m^6^A increase	METTL3	Up- *BCL-XS*, *NCOR*	GBM cell lines	[76]
	m^6^A increase	METTL3	Down- *OPTN*	GSCs	[77]
	m^6^A decrease	ALKBH5	Up- *LOC100507424* Up- *FOXM1*	GSCs	[78]
	m^6^A increase	FTO	Down- *ATF5*	GSCs	[81]
	m^6^A increase	ALKHB	Up- *PDGFR-B*	GSCs	[82]
	m^6^A decrease	ALKBH5	-	GSCs	[84]
	-	YTHDF2	Down- *CDKN2B*	(Li–Fraumeni syndrome astrocytes)	[85]
	m^6^A decrease	METTLE	Down- *MGMT*	GSCs	[86]
Pseudouridine	ψ decrease at tRNA	PUS7	Up- TYK2-IFN	GSCs	[88]
	ψ increase	DKC1	Up- Telomerase	GBM cell lines	[90]
m^5^C	m^5^C increase	NSUN4	Up- *CDC42*, PI3K-AKT	GBM cell lines, In vivo model	[92]
	m^5^C increase	NSUN2	Up- *VEGFR2*	GBM endothelial cells	[93]
m^5^C/hm^5^C	hm^5^C increase via m^5^C	NSUN5/TET2	Down- *CTNNB*	Tumor-associated macrophages, In vivo model	[94]
m^5^C	m^5^C decrease	NSUN5	28s rRNA (protein translation depletion)	GBM cell lines, In vivo model	[95]
A-to-I editing	Inosine increase	ADAR1	Up- *GM2A*	GSCs In vivo model	[96]
	Unedition	ADAR2	miR-589-3P unedited (oncogenic), Up- *ADAM12*	Glioma cell lines	[99]
	Edition (Inosine increase)	ADAR2	miR-589-3P edited (non-oncogenic), Up- *PCDH9*		
	Inosine increase	ADARB1	miR-376, Down- *RAP2A*, Up- *AMFR*	Glioma cell lines In vivo model	[100]

Up-: Upregulation. Down-: Downregulation.

Overall, an improper interpretation of the signals encoded by RNA modifications in various molecules can lead to the emergence of neoplastic behavior in brain tissue, ultimately resulting in the development of gliomas. However, as seen for other tumorigenic pathways, the contribution of each mechanism to tumor development is intricate and challenging to define. It is likely that the interaction among these dysregulations provides an evolutionary advantage to tumor cells, promoting their proliferation.

## 4. Targeting Epitranscriptomic Modifiers

Similarly to studies that leverage DNA methylation as a biomarker for specific tumors, epitranscriptomic alterations represent a promising avenue for developing diagnostic and prognostic strategies [98,101,102] as well as potential targets for clinical treatments. The editing of *Alu* repeats has been reported to have prognostic value in GBM in a gender-specific manner, with highly edited sequences correlating with better prognosis in males but showing the opposite association in females [98]. m^6^A distribution in lncRNAs was also suggested as a possible prognostic marker in low-grade gliomas, as well as the expression of its regulators [103,104]. It has also been reported that m^6^A patterns are associated with immune infiltration, enabling the identification of distinct immune invasion phenotypes. These patterns or signatures could predict key factors such as inflammation, prognosis, mutation burden, and, potentially, a better response to immunotherapy [105], a finding also reported for m^5^C modifications in the mRNA of specific genes [106]. As discussed in the previous sections, data derived from RNA modifications can be considered as potential clinical information by targeting genes associated with these modifications. In the study by Zhou and colleagues, a signature that predicts the outcome of low-grade gliomas based on diverse RNA modification regulators was developed [107]. Several studies have combined distinct RNA modification patterns to establish signatures that define tumor subtypes, with different clinical implications. Many of these studies generate a risk score based on various parameters, including RNA modifications and clinical data, to classify patients. This approach is particularly valuable, as some relationships reveal distinct immune infiltration profiles or the expression of specific genes, offering insights into potential therapeutic combinations, including immunotherapy, based on patient status. While these studies rely on multivariate analyses of patient cohorts, the next step could involve using 3D organoids derived from patient cells and future in vivo models to validate these signatures and assess their clinical applicability.

The emphasis on investigating the role of m^6^A across several processes has positioned the inhibition of readers—METTL3, METTL14, and WTAP—and erasers—FTO and ALKBH5—as the primary strategy in the development of therapeutic targets [102]. On the one hand, the catalytic inhibition of METTL3 is expected to decrease the levels of m^6^A and contribute to reducing tumorigenesis. In acute myeloid leukemia, the METTL3 inhibitor STM2457 reduces m^6^A levels in specific cancer-related genes, enhances their expression, and ultimately extends survival in AML mouse models [108]. Years later, STM3006 was presented as an improved version of STM2457 that generates double-strand RNA structures, generated as a consequence of low levels of m^6^A after METTL3 inhibition, activating the IFN response [109]. Parallelly, STC-15, another METTL3 inhibitor analogous to STM3006, contributes to the innate immune response and reduces tumor growth via T cell checkpoint blockade treatments, such as anti-PD-1 therapy. STC-15 was the first-in-class RNA-modifying enzyme to enter clinical development (Phase I, NCT05584111) and it has also demonstrated tolerance and signs of clinical activity in patients with advanced malignancies [110]. Other inhibitors such as UZH1a [111] and its improved version UZH2 [112], have also been reported in recent years, though their testing has been limited to in vitro assays. Additionally, interference with recognition motifs of YTHDF reader proteins can mitigate the effects of elevated m^6^A levels. For instance, the small molecule Ebselen covalently binds to the YTH domain, disrupting its interaction with m^6^A [113]. Studies on YTHDF proteins showed that their inhibition has antitumor effects. DF-A7, a small-molecule inhibitor, degrades YTHDF2 and boosts CD8^+^ T cell-mediated cytotoxicity, with potential for combination with anti-PD-L1 or PD-1 therapies, as observed with the METTL3 inhibitor STC-15 [114]. In addition, YTHDF2 inhibition associated with the DC-Y13-27 compound has been shown to enhance immune responses and also represents a promising target for combination therapy with anti-PD-L1 and radiotherapy in myeloid tumors [115]. On the other hand, other studies have focused on targeting m^6^A eraser proteins, such as FTO and ALKBH5, as a strategy to stabilize m^6^A levels. Meclofenamic acid has been proven to be a specific inhibitor of FTO compared to ALKBH5, showing increased m^6^A levels and reduced proliferation in acute myeloid leukemia and GSCs [116,117]. Similarly, the 18097 molecule specifically binds to the active site of FTO, elevates m^6^A levels, and suppresses tumor growth in vivo in a xenograft model of lung cancer [118]. Other FTO inhibitors reported were obtained after in silico virtual screenings, including oxetanyl inhibitors in GBM cells [119] and CS1 and CS2 in acute myeloid leukemia cells [120]. Targeting ALKH5B has been related with improvements in immunotherapy. *ZDHHC3* downregulation is related to an m^6^A increase when ALKH5B is inhibited, which suppresses *PD-L1* expression, making tumor cells a potential target for PD-1 and PD-L1 inhibitors [69].

Pharmacological interventions and surgery are key for treating gliomas, but the blood–brain barrier (BBB) limits drug delivery. While some GBM regions show BBB breakdown, infiltrated non-tumor areas remain intact, creating an uneven drug distribution [121]. RNA-based therapeutics face similar challenges. Delivery strategies include invasive (brain stimulation, direct injection, intracerebral devices) and non-invasive approaches (nanoparticles, liposomes, exosomes, cell-mediated delivery, neurotropic viruses, cell-penetrating peptides, and focused ultrasound to modulate BBB permeability) [122]. Although it is early to tackle this issue with RNA-modifying drugs for gliomas, protein inhibitors must be tailored to specific delivery systems to enhance effectiveness and minimize potential damage to non-tumor cells. Approximately 97% of clinical trials fail to gain FDA approval due to issues with compound efficacy and toxicity [123]. Off-target effects are commonly observed in clinical trials and can activate unexpected pathways and jeopardize patient health. In epi-drug development, targeting epigenetic or epitranscriptomic imbalances, caution is needed when altering DNA or RNA modification patterns, since this may unintentionally disrupt other transcripts and amplify system imbalances [124]. On a genomic scale, epigenetic or epitranscriptomic targets must be specifically addressed to avoid a severe impact across all genes or transcripts. In this vein, systems like sequence-specific DNA-binding domains (DBDs) can direct epigenetic drugs to specific genomic locations to alter DNA [125] or RNA marks.

However, despite recent advancements, it remains too early to foresee the clinical application of these inhibitors in gliomas or other neoplasms. Currently, most research is centered on developing high-throughput screenings to identify additional small molecules that target epitranscriptomic modifiers, like the examples mentioned above [126,127,128]. An exception is shown with STC-15, which is being checked in terms of dose escalation safety. The coming years will be crucial for understanding the potential of new drugs targeting this machinery and evaluating their effects in future clinical trials. Efforts to design inhibitors targeting epitranscriptomic modifiers will not only facilitate the development of other new therapeutic approaches but also provide valuable insights into the functional roles of these proteins. Such advancements are expected to drive innovation in drug discovery and deepen our understanding of the complex mechanisms underlying epitranscriptomic regulation.

## 5. Conclusions

Epitranscriptomics has emerged as a critical regulatory network essential for maintaining cellular and tissue homeostasis. While the role of RNA modifications in stabilizing RNA molecules and ensuring their proper function has been recognized for years, recent studies underscore the delicate balance maintained by the network of writers, readers, and erasers involved in epitranscriptomic regulation. This balance is essential to prevent the emergence of aberrant signals that could activate unintended pathways. As discussed, alterations in RNA modification patterns, often driven by changes in protein expression, can disrupt this balance, leading to the inappropriate activation or suppression of protein synthesis. Such dysregulation may drive tumorigenic effects, including uncontrolled cell proliferation, invasion, or self-renewal, thus highlighting the oncogenic potential of epitranscriptomic alterations. While specific inhibition of epitranscriptomic regulators is still in its infancy, the knowledge gained over the past decade offers a promising foundation for identifying new tumor vulnerabilities that could be targeted through therapeutic interventions.

## Figures and Tables

**Figure 1 cancers-17-00578-f001:**
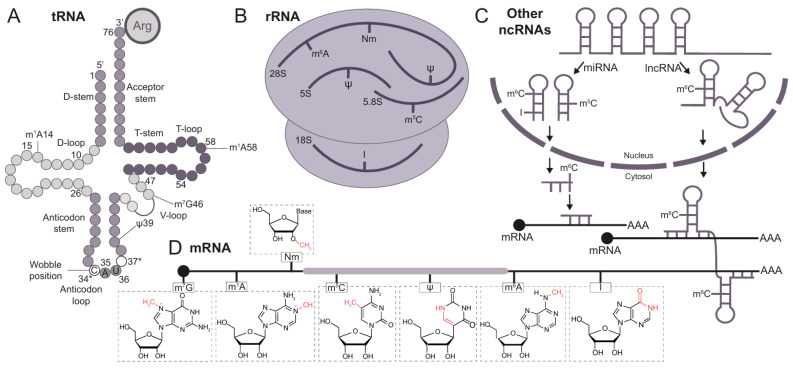
Epitranscriptomic modifications across distinct RNA species. RNA modifications are found in all RNA molecules, both coding and non-coding. (**A**). Each circle represents an individual base in tRNAs. The three-dimensional structure of tRNA is organized into distinct loops. The anticodon loop is the region that interacts with the mRNA, and the surrounding bases are susceptible to a wide range of modifications (marked with *). The 34th position is designated as the wobble position, providing high flexibility when tRNAs recognize a specific codon. (**B**). Distinct rRNAs bound to ribosomes are targets of various modifications. (**C**). Small non-coding RNAs undergo cleavage steps to generate final molecules of different sizes. The stabilization of these molecules, which are bound to distinct targets (e.g., mRNAs), is sometimes dependent on specific post-transcriptional modifications, influencing their stabilization or degradation. (**D**). Post-transcriptional modifications in mRNAs also contribute to their stabilization by activating or inhibiting degradation pathways. A list of the different modifications mentioned is as follows: m^7^G: N7-methylguanosine; m^1^A: N^1^-methyladenosine; Nm: 2′-O-methylation; m^5^C: 5-methylcytosine; Ψ (pseudouridine); m^6^A: N6-methyladenosine; and I: inosine.

**Figure 2 cancers-17-00578-f002:**
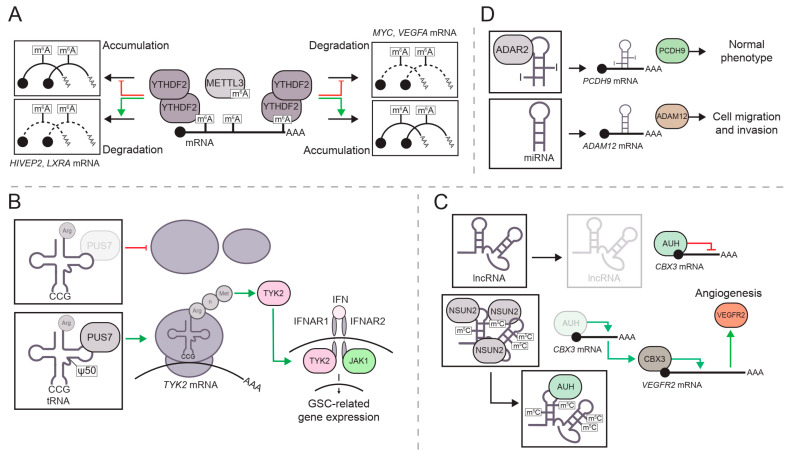
Epitranscriptomic modifications across distinct RNA species. Post-transcriptional modifications can be involved in gene expression switching by modulating different pathways and mechanisms. (**A**). YTHDF2 can activate or deactivate distinct pathways. By binding to m^6^A, YTHDF2 can promote the accumulation and translation of *MYC* and *VEGF* mRNA, while facilitating the degradation of other mRNAs, such as *HIVEP2* and *LXRA*, thereby enhancing the glioma phenotype. (**B**). Overexpression of *PUS7* elevates pseudouridine levels at the 50th position of arginine tRNAs, which are involved in the translation of TYK2, an intermediary in IFN pathways that contributes to oncogene expression. (**C**). In the absence of m^5^A, lncRNA *LINC00324* is degraded, and AUH inhibits *CBX3* expression. The accumulation of NSUN2 increases m^5^C levels in *LINC00324*, allowing it to compete with *CBX3* mRNA for AUH binding. When AUH is absent, *CBX3* expression is activated, leading to elevated levels of *VEGFR2*, contributing to angiogenesis in GBM endothelial cells. (**D**). Distinct patterns of A-to-I modification can alter the specificity of miRNAs. Under physiological conditions, ADAR2 edits miR-589-3P, which enhances *PCDH9* expression and contributing to cell homeostasis. In the GBM phenotype, low *ADAR2* expression results in unedited miR-589-3P, which switches its target to *ADAM12* mRNA, a metalloproteinase involved in cell migration and invasion, ultimately contributing to the GBM phenotype.

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
