# Peer review of "Epitranscriptomics in the Glioma Context: A Brief Overview"

_cancers, 2025, doi:10.3390/cancers17040578_

Round 1

Reviewer 1 Report

Comments and Suggestions for Authors

In the work, „Epitranscriptomics in the glioma context: a brief overview“, the authors Santamarina-Ojeda et al. present a brief, however very detailed review of the different mechanisms of RNA epigenetics; they give an impressive overview on basic mechanisms, and regarding clinicopathological aspects, they focus on gliomas, as indicated in the title. The work is concise and well written, it brings an enormous lot of information together on a short space, and it contains a lot of references that are relevant in the field.

Most oft he paper concerns general mechanisms, and the particular aspects of gliomas are only in the focus of two subheadings. It becomes clear that there is a very high number of epigenetic modifications of RNAs that influence malignancy; and that for gliomas, in particular glioblastomas, there still is a lot unknown. This becomes even more clear regarding possible options of therapeutic interventions – however, some interesting speculations are presented, which is quite amazing.

The authors demonstrate possible interactions between IDH alterations – a phenomenon commonly seen in glioblastomas – and the RNA landscape, however  other molecules like EGFR, PTEN, TP53, MGMT, and others are not mentioned, and for PDGFR only one reference is cited. I wonder whether any relations have been described here or may be speculated about?

Anyway, the work contains an enormous lot of information and is well written, thus, I would recommend its publication.

Author Response

Comments 1: “The authors demonstrate possible interactions between IDH alterations – a phenomenon commonly seen in glioblastomas – and the RNA landscape, however  other molecules like EGFR, PTEN, TP53, MGMT, and others are not mentioned, and for PDGFR only one reference is cited. I wonder whether any relations have been described here or may be speculated about?”

Response 1: We sincerely appreciate your thoughtful review and valuable suggestions/questions. As reviewer #1 highlights, genetic alterations in genes such as TERT, IDH1, EGFR, PTEN, TP53, MGMT, CDKN2B and PDGFR play a crucial role on gliomagenesis and serve as biomarker for classifying patients into distinct prognostic groups and, indeed, distinct tumors. Given that mutations and chromosomal amplifications (genetic/DNA level alterations) are primary drivers of their dysregulation, we initially chose not to discuss epitranscriptomic modifications in these genes directly but rather in the context of other pathways. However, we acknowledge that including this perspective enhances the relevance and impact of our paper. In response to Reviewer #1's suggestion, we have now incorporated details on lines 235-255 regarding the influence of epitranscriptomics on these genes.

Reviewer 2 Report

Comments and Suggestions for Authors

This review provides a fresh perspective by focusing on the emerging role of epitranscriptomic modifications in glioma biology, an area still in its nascent stage. It highlights the intricate interplay between RNA modifications and tumorigenesis, offering insights into their potential as diagnostic markers and therapeutic targets. By bridging gaps in understanding the biological implications of RNA modifications in gliomas, this work sets the stage for advancing precision medicine in neuro-oncology. The work need revisions as suggested in comments.

1.      The Glioma information needs to update with recent work on progression and prediction with recent work of 2024 like https://doi.org/10.1111/cns.14489, https://doi.org/10.18632/aging.205952, http://dx.doi.org/10.2174/1568009623666230817102104 etc., where ever applicable. 

2.      Include more detailed discussions on how specific RNA modifications, such as N6-methyladenosine (m6A) or pseudouridine, influence glioma biology, focusing on key molecular pathways or signaling mechanisms.

3.      Add the latest research and breakthroughs in epitranscriptomics, particularly studies published in the last two years, to provide a comprehensive and up-to-date overview. (At present 3-4 work from 2024 and 9-10 from 2023, expected all recent data to its best).

4.      Please elaborate on the therapeutic implications of targeting RNA modifications, including ongoing clinical trials, challenges in drug delivery, and potential off-target effects.

5.      It is highly appreciated if comparison of epitranscriptomic mechanisms in gliomas versus other cancers to contextualize the uniqueness or commonalities of these modifications.

6.      The two figures are impressive (Check for originality, if reprinted or re-concept please cite them) however, addition of few research study images will increase its values.

7.      Review lacks the tabular data. It's highly recommended to add a table that lists key RNA modifications and their roles in glioma development.

8.      Expecting, a paragraph on potential concerns related to targeting epitranscriptomic mechanisms, such as systemic toxicity or unintended effects on normal cells.

9.   Check the document for ref style. Instead of [2,3,4], it should be [2-4]. Check the text carefully for such errors. 

Author Response

Comments 1: The Glioma information needs to update with recent work on progression and prediction with recent work of 2024 like https://doi.org/10.1111/cns.14489, https://doi.org/10.18632/aging.205952, 

http://dx.doi.org/10.2174/1568009623666230817102104 etc., where ever applicable. 

Response 1: We have carefully reviewed the information regarding glioma and glioblastoma. To address this, we have incorporated details on how adult glioma patients are classified (section "Role of RNA modifications in gliomagenesis", lines 165–179) and updated information on the role of various RNA modifications as potential prognostic biomarkers (section "Targeting epitranscriptomic modifiers", lines 139–157). While the suggested papers provide valuable insights into the prognostic relevance of certain miRNAs, lncRNAs, and specific genes, we have opted to maintain the focus of this review on epitranscriptomic mechanisms. This decision was made to prevent expanding the scope towards non-epitranscriptomic regulatory mechanisms, such as non-coding RNAs or genetic regulation.

---

Comment 2: Include more detailed discussions on how specific RNA modifications, such as N6-methyladenosine (m6A) or pseudouridine, influence glioma biology, focusing on key molecular pathways or signaling mechanisms.

Response 2: We have expanded our discussion by including more detailed examples of how RNA modifications, specifically m6A and pseudouridine, influence gliomagenesis through key molecular pathways and signaling mechanisms. We have also included a part where we mention potential epitranscriptomic disturbancies in canonical genes such as IDH1, EGFR, TP53, MGMT, CDKN2B and PDGFR. As m6A is the most extensively studied RNA modification to date, the number of references to this mark aligns with the key studies published on this topic. These additions can be found in the section "Role of RNA Modifications in Gliomagenesis", on lines 235-255 and 265-268.

---

Comment 3: Add the latest research and breakthroughs in epitranscriptomics, particularly studies published in the last two years, to provide a comprehensive and up-to-date overview. (At present 3-4 work from 2024 and 9-10 from 2023, expected all recent data to its best).

Response 3: We have updated 8 references with the latest discoveries in epitranscriptomics in the “Description of the key RNA modifications” section, which are highlighted in red for easy identification (16, 17, 27, 28, 31, 48, 53, 55). The references that have not been updated meet the following criteria: 1) They represent meaningful discoveries published in high-impact journals (e.g., Nature, Cell, Genome Research, Nucleic Acids Research); 2) There have been no significant updates in the last few years; 3) They are the first publications on specific topics (e.g., first identification of methylated RNA, as referenced in 51 and 52).

---

Comment 4: Please elaborate on the therapeutic implications of targeting RNA modifications, including ongoing clinical trials, challenges in drug delivery, and potential off-target effects.

Response 4: We have incorporated more detailed and up-to-date information on recent advances in targeting epitranscriptomic modifiers. Currently, the only clinical trial (Phase I) involves STORM Therapeutics’ METTL3 inhibitor (STC-15), which has shown efficacy in preclinical models. However, enrollment in this trial requires histologic or cytologic confirmation of advanced malignancies that have failed standard treatments, without specific inclusion of glioma tumors. Other approaches to targeting epitranscriptomic modifiers remain in early in vitro stages and will require validation in preclinical models before progressing to clinical trials. Additionally, we have included information regarding drug delivery system posibilities in the future as well as the potential off-target effects using these drugs. In response to Reviewer #2, we have expanded the discussion on therapeutic implications, drug delivery, and off-target effects in the section "Targeting Epitranscriptomic Modifiers", lines 344-398.

---

Comment 5: It is highly appreciated if comparison of epitranscriptomic mechanisms in gliomas versus other cancers to contextualize the uniqueness or commonalities of these modifications.

Response 5: Gliomas, particularly glioblastomas, and hematologic malignancies have been extensively studied from an epigenetic and epitranscriptomic perspective. The complexity of the central nervous system, with its diverse cell types (astrocytes, neurons, microglia, oligodendrocytes, endothelial cells, etc.), contributes to the heterogeneity of gliomas. Unlike other tumors that rely on a single mutation driving a specific pathway —making targeted therapies more straightforward— high-grade gliomas exhibit vast heterogeneity, necessitating a multifaceted approach to fully understand their biology and develop effective therapeutic strategies. In this context, studies on additional regulatory layers, such as epigenetics and epitranscriptomics, have identified multiple dysregulated pathways that may serve as therapeutic targets, diagnostic biomarkers, or prognostic indicators. In response to Reviewer #2’s suggestion, we have incorporated this discussion in the section Role of RNA Modifications in Gliomagenesis, lines 155-179.

---

Comment 6: The two figures are impressive (Check for originality, if reprinted or re-concept please cite them) however, addition of few research study images will increase its values.

Response 6: We sincerely appreciate your thoughtful review and positive feedback on the figures. We confirm that both figures are entirely original to this paper. Regarding the addition of research study images, we would greatly appreciate further clarification on the specific type of images the reviewer envisions. This additional guidance will help us better address the suggestion and enhance the manuscript accordingly.

--

Comment 7: Review lacks the tabular data. It's highly recommended to add a table that lists key RNA modifications and their roles in glioma development.

Response 7: We fully agree with this observation and recognize the value of providing more clarity on the various RNA modifications in gliomas. In response, we have added a table (Table 1) summarizing the key RNA modifications and their roles in glioma development, as mentioned throughout the text.

---

Comment 8: Expecting, a paragraph on potential concerns related to targeting epitranscriptomic mechanisms, such as systemic toxicity or unintended effects on normal cells.

Response 8: We have addressed this concern in the “Targeting epitranscriptomic modifiers” section, lines 344-398, as mentioned in Response #4. In these lines, we discuss potential concerns related to systemic toxicity and unintended effects on normal cells when targeting epitranscriptomic mechanisms.

---

Comment 9: Check the document for ref style. Instead of [2,3,4], it should be [2-4]. Check the text carefully for such errors. 

Response 9: We have carefully reviewed and revised the reference style throughout the document, changing instances of [2,3,4] to [2-4] and addressing other similar formatting errors.

Reviewer 3 Report

Comments and Suggestions for Authors

This paper describes a short review about epitranscriptomics in the glioma context. The present structure of the paper is not suitable for a revision paper for cancers. Indeed, the following is not appropriate:

- The type of the paper is Review and not Article

- The paper contains only two main sections: Introduction and Conclusions. 

- The introduction should contain a last paragraph where the objective of the revision is described and the time period of the revision must be clearly stated. Also, previous revisions about the same subject must be cited and the novelty of the present revision described.

- The first two/three pages of the revision is about the chemical modifications of the different RNA types. This is probably described in previously published revisions and could be replaced by the citation of these revisions.

- The following four pages focus on the subject of the present revision. This discussion should be extended because this is the main objective of the paper. Also, tables and figures (artwork) must be added to this discussion. 

- About 11 pages of bibliography (in a different format and bigger size of the letter) with a different format must be corrected. 

Author Response

We greatly appreciate your thoughtful and constructive suggestions. Your comments have improved the clarity and quality of the manuscript. We have carefully considered each suggestions and we have made the revisions accordingly.

Comment 1: The type of the paper is Review and not Article

Response 1: This paper was originally submitted as a review, in accordance with its intended format. We have revised it to address any potential errors.

---

Comments 2: The paper contains only two main sections: Introduction and Conclusions. 

Response 2: We have updated the heading to clearly distinguish it as a different format. The revised submission will include the following sections: 1. Introduction; 2. Description of the Key RNA Modifications; 2.1. Chemical Modifications at tRNAs; 2.2. Chemical Modifications at rRNAs; 2.3. Chemical Modifications at Other ncRNAs; 2.4. Chemical Modifications at mRNAs; 3. Role of RNA Modifications in Gliomagenesis; 4. Targeting Epitranscriptomic Modifiers; and 5. Conclusions.

---

Comments 3: The introduction should contain a last paragraph where the objective of the revision is described and the time period of the revision must be clearly stated. Also, previous revisions about the same subject must be cited and the novelty of the present revision described.

Response 3: We have incorporated these suggestions into the “Introduction” section and separated the final paragraph to better highlight its content. Additionally, we have cited previous reviews on the same topic to provide context, and the references have been thoroughly reviewed (lines 163-165). Regarding similar revisions, we have emphasized the novelty of our review. The most recent reviews, from mid-2023, share a similar structure with this manuscript and address the same issues in the field. However, due to the rapid growth of the epitranscriptomics field, we have included novel studies from the past year/months across different sections of this review, as well as research on new epitranscriptomic drugs. The aim of this review is to provide a concise yet comprehensive overview of the main RNA modifications, and we believe that all reviews in this area are complementary and contribute to enriching the field. We have included similar comments in lines 62-69.

---

Comments 4: The first two/three pages of the revision is about the chemical modifications of the different RNA types. This is probably described in previously published revisions and could be replaced by the citation of these revisions.

Response 4: In line with several reviews on epitranscriptomics and related fields such as epigenetics, the initial sections of these papers often provide an overview of the key regulatory players. Our goal was to create a review accessible to both experts and those less familiar with the field. Including this background information not only facilitates a broader readership but also enhances the potential for future citations—whether by researchers referencing different RNA modifications and their roles or by those seeking a comprehensive introduction to the topic. Other reviewed consulted were properly cited when corresponds.

---

Comments 5: The following four pages focus on the subject of the present revision. This discussion should be extended because this is the main objective of the paper. Also, tables and figures (artwork) must be added to this discussion. 

Response 5: We have expanded the sections "Role of RNA Modifications in Gliomagenesis" and "Targeting Epitranscriptomic Modifiers" to include more examples and provide a broader perspective on the current state of epitranscriptomic therapeutics, as well as the challenges these face in reaching clinical application. Additionally, we have added a table (Table 1) summarizing the key pathways altered in gliomas. The two figures presented now offer visual information on the various RNA modifications across different RNA species and highlight examples of how these modifications may influence gliomagenesis.

---

Comments 6: About 11 pages of bibliography (in a different format and bigger size of the letter) with a different format must be corrected. 

Response 6: We have included 128 bibliographic references in order to provide a complete revision of this issue. In the initial submission, we have not adopted any format for the bibliographic references, but the format should be changed at time to publication.

Round 2

Reviewer 2 Report

Comments and Suggestions for Authors

The revision is satisfactory. 

Thank you 

Reviewer 3 Report

Comments and Suggestions for Authors

The authors have markedly improved this revision.